

# Completeness of the Bethe states for the rational, spin-1/2 Richardson–Gaudin system

**Jon Links**

School of Mathematics and Physics, The University of Queensland, 4072 Australia

[jrl@maths.uq.edu.au](mailto:jrl@maths.uq.edu.au)

## Abstract

**Establishing the completeness of a Bethe Ansatz solution for an exactly solved model is a perennial challenge, which is typically approached on a case by case basis. For the rational, spin-1/2 Richardson–Gaudin system it will be argued that, for generic values of the system's coupling parameters, the Bethe states are complete. This method does not depend on knowledge of the distribution of Bethe roots, such as a string hypothesis, and is generalisable to a wider class of systems.**

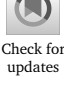

doi:10.21468/SciPostPhys.3.1.007

## 1 Introduction

Since Bethe first introduced the notion of an Ansatz approach through his study of the Heisenberg $XXX$ chain [1], a recurring question is whether a particular Bethe Ansatz solution is complete, i.e. provides a basis of eigenstates. For the $XXX$ chain this is a problem which has

attracted significant attention (see [2] for a historical overview), and one which has continued to produce research results up until recent times [3–7]. Similar analyses have been performed for other models, e.g. [8–11]. In many of these studies steps are undertaken to first identify that solutions of the Bethe Ansatz equations have some regular structure, such as that of a *string hypothesis*, and then a counting argument is applied to determine the number of such solutions. However this approach is problematic due to the intricate nature of the strings [12]. Furthermore it is challenged by the fact that there are *singular* and/or *exceptional* [3, 6, 7, 13] solutions which, roughly speaking, are associated with divergent Bethe roots and/or Bethe vectors that may or may not be regularisable. These features complicate the counting process.

The rational, spin-1/2 Richardson–Gaudin system is closely related to the *XXX* chain in the sense that it can be derived in the *quasi-classical* limit of the Quantum Inverse Scattering Method based on the rational six-vertex solution of the Yang-Baxter equation [14]. It will be shown below that, for generic values of the system's coupling parameters, completeness of the Bethe Ansatz solution can be deduced without a string hypothesis. Moreover, there are no spurious solutions. The approach taken is very much influenced by the development of the *off-diagonal* Bethe Ansatz method based on functional relations [15].

## 1.1 The rational, spin-1/2 Richardson–Gaudin system

Before describing the rational, spin-1/2 Richardson–Gaudin system, some notational conventions need to be fixed. Let $\{|+\rangle, |-\rangle\}$ denote a basis for $\mathbb{C}^2$ and let $\{I, \sigma^z, \sigma^+, \sigma^-\}$ denote a basis for $\mathrm{End}(\mathbb{C}^2)$ with the actions

$$
\begin{aligned}
I|\pm\rangle &= |\pm\rangle, & \sigma^z|\pm\rangle &= \pm|\pm\rangle, \\
\sigma^\pm|\mp\rangle &= |\pm\rangle, & \sigma^\pm|\pm\rangle &= 0.
\end{aligned}
$$

For arbitrary $\alpha \in \mathbb{R}$, and pairwise-distinct real parameters $\{z_j : j = 1, ..., L\}$, the rational, spin-1/2 Richardson–Gaudin system is described by the self-adjoint operators $T_j \in \mathrm{End}(\mathbb{C}^2)^{\otimes L}$ of the form

$$
T_j = \alpha \sigma_j^z + \sum_{k \neq j}^{L} \frac{\mathscr{P}_{jk} - I}{z_j - z_k}, \qquad j = 1, ..., L, \tag{1}
$$

which are mutually commuting, viz

$$
\left[ T_j, T_k \right] = 0 \qquad \forall\, j, k = 1, ..., L.
$$

Above, $\mathscr{P} \in \mathrm{End}(\mathbb{C}^2)^{\otimes 2}$ is the permutation operator satisfying

$$
\mathscr{P}(\mathbf{x} \otimes \mathbf{y}) = \mathbf{y} \otimes \mathbf{x}, \qquad \forall\, \mathbf{x}, \mathbf{y} \in \mathbb{C}^2.
$$

The subscripts associated with operators appearing in the right-hand side of (1) denote the tensor components within $(\mathbb{C}^2)^{\otimes L}$ on which each of these operators act.

Due to the properties of being self-adjoint and mutually commuting the set $\{T_j : j = 1, ..., L\}$ is simultaneously diagonalisable, which is described in terms of a Bethe Ansatz solution. For $\alpha = 0$, the operators (1) first appeared in the work of Gaudin in 1976 [16], along with the Bethe Ansatz solution. Commutativity of the operators for $\alpha \neq 0$ was established by Sklyanin using the Quantum Inverse Scattering Method in the quasi-classical limit [14]. Independently, the same operators were constructed in [17] and shown to commute with the *s*-wave pairing Hamiltonian which was solved by Richardson much earlier in 1963 [18]. The concrete connection between Richardson's solution and Quantum Inverse Scattering Method results was made in 2002 [19, 20].

To describe the Bethe Ansatz solution of the system, the eigenvalue of each conserved operator $T_j$ on a simultaneous eigenstate takes the form

$$\lambda_j = \alpha - \sum_{m=1}^{M} \frac{1}{z_j - v_m}, \tag{2}$$

where the parameters $\{v_m : m = 1, ..., M\}$ satisfy the Bethe Ansatz equations

$$2\alpha + \sum_{j=1}^{L} \frac{1}{v_m - z_j} = \sum_{n \neq m}^{M} \frac{2}{v_m - v_n}, \qquad m = 1, ..., M. \tag{3}$$

For $\alpha \neq 0$ the operator

$$S = \frac{1}{2\alpha} \sum_{j=1}^{L} T_j$$
$$= \frac{1}{2} \sum_{j=1}^{L} \sigma_j^z,$$

has eigenvalues $s$ with integer spacing, belonging to the set

$$\left\{ \frac{L}{2}, \frac{L}{2} - 1, ... - \frac{L}{2} + 1, -\frac{L}{2} \right\}.$$

Thus the operator $S$ is a $u(1)$ invariant which partitions $(\mathbb{C}^2)^{\otimes L}$ into sectors associated with each $s$. Specifically,

$$(\mathbb{C}^2)^{\otimes L} = \bigoplus_{s=-L/2}^{L/2} V_s \tag{4}$$

such that $S\mathbf{w} = s\mathbf{w}$ for $\mathbf{w} \in V_s$. From the Bethe Ansatz solution (3) it is found that

$$\frac{1}{2\alpha} \sum_{j=1}^{L} \lambda_j = \frac{L}{2} - \frac{1}{2\alpha} \sum_{j=1}^{L} \sum_{m=1}^{M} \frac{1}{z_j - v_m}$$
$$= \frac{L}{2} - M.$$

which establishes the relationship $s = L/2 - M$. It is worth highlighting that the inclusion of a non-zero $\alpha$ dependence in (1) is akin to twisting the periodic boundary conditions in the $XXX$ chain. In the limit $\alpha = 0$ the symmetry of the system is enhanced from $u(1)$ to $gl(2)$. Further details may be found in [14, 19, 20].

Explicit expressions for the eigenstates associated with each solution of (3), which will be referred to as *Bethe states*, have the form [14, 19, 20]

$$|\Phi\rangle = \prod_{k=1}^{M} \left( \sum_{j=1}^{L} \frac{1}{v_k - z_j} \sigma_j^- \right) |0\rangle, \tag{5}$$

where $|0\rangle = |+\rangle^{\otimes L}$ is known as the pseudo-vacuum state.

A solution set of (3) with $M$ roots provides a candidate for determining a simultaneous eigenstate of the $T_j$ associated with the sector for which $S$ has eigenvalue $s = L/2 - M$. It is possible that such a solution of (3) does not correspond to a simultaneous eigenstate, in cases where (5) vanishes. In the literature this is commonly known as a *spurious* solution.

One aim of this work is to establish that there are no spurious solutions of (3). Moreover, it will be described how a complete set of eigenstates is constructed from solutions of (3) in the case when $\alpha \neq 0$. It appears that there is not much published literature on this problem for this integrable system, other than some studies for the $\alpha = 0$ case [21, 22]. In that instance, counting of states requires the identification of singular solutions associated with divergent Bethe roots, which is avoided here. The problem for the $gl(n)$ generalisation, still with $\alpha = 0$, has been studied in [23]. There the analysis was conducted by making use of the separation of variables method, which was first developed by Sklyanin for the $gl(2)$ case [14].

The arguments will be presented as follows. Sect. 2 establishes a set of operator identities, from which the Bethe Ansatz equations (3) are obtained through the associated operator eigenvalue identities. In Sect. 3 attention turns towards the construction of the corresponding Bethe states. Then in Sect. 4 an argument is presented asserting that, under certain mild conditions, all eigenstates can be constructed through the above procedure. Concluding remarks are provided in Sect. 5.

## 2 Operator identities and the Bethe Ansatz equations

It was observed in [24, 25] that a different form of the Bethe Ansatz solution can be presented solely in terms of the variables $\{\lambda_j : j = 1, ..., L\}$, rather than the variables $\{\nu_k : k = 1, .., M\}$, which has a quadratic form:

$$\lambda_j^2 = \alpha^2 - \sum_{k \neq j}^{L} \frac{\lambda_j - \lambda_k}{z_j - z_k}, \qquad j = 1, ..., L. \tag{6}$$

This form of solution has recently been proposed [25], along with those for similar systems [25–27], as an alternative for both numerical computation of the spectra for the commuting operators and for the development of correlation functions.

Let $W_\lambda$ denote the simultaneous eigenspace associated with the eigenvalues $\{\lambda_j : j = 1, , ..., L\}$.

**Proposition 1**: The following operator identities hold,

$$T_j^2 = \alpha^2 I - \sum_{k \neq j}^{L} \frac{T_j - T_k}{z_j - z_k}, \qquad j = 1, ..., L. \tag{7}$$

*Proof.* Using the key relations which hold for $j \neq k \neq l \neq j$:[1]

$$\mathscr{P}_{jk}\mathscr{P}_{jl} + \mathscr{P}_{jl}\mathscr{P}_{jk} = \mathscr{P}_{jk} + \mathscr{P}_{jl} + \mathscr{P}_{kl} - I, \tag{8}$$
$$\sigma_j^z \mathscr{P}_{jk} + \mathscr{P}_{jk}\sigma_j^z = \sigma_j^z + \sigma_k^z$$

---

[1]Note that (8) only holds in $\mathrm{End}(\mathbb{C}^2)^{\otimes L}$ and not in $\mathrm{End}(\mathbb{C}^n)^{\otimes L}$ generally.

a direct calculation verifies (7):

$$
\begin{aligned}
T_j^2 &= \alpha^2 I + \alpha \sum_{k \neq j}^{L} \frac{\sigma_j^z(\mathscr{P}_{jk} - I) + (\mathscr{P}_{jk} - I)\sigma_j^z}{z_j - z_k} + \sum_{k \neq j}^{L} \frac{\mathscr{P}_{jk} - I}{z_j - z_k} \sum_{l \neq j}^{L} \frac{\mathscr{P}_{jl} - I}{z_j - z_l} \\
&= \alpha^2 I - \alpha \sum_{k \neq j}^{L} \frac{\sigma_j^z - \sigma_k^z}{z_j - z_k} + \frac{1}{2} \sum_{k \neq j}^{L} \sum_{l \neq k, j}^{L} \frac{\mathscr{P}_{kl} - \mathscr{P}_{jk} - \mathscr{P}_{jl} + I}{(z_j - z_k)(z_j - z_l)} - 2 \sum_{k \neq j}^{L} \frac{\mathscr{P}_{jk} - I}{(z_j - z_k)^2} \\
&= \alpha^2 I - \sum_{k \neq j}^{L} \frac{T_j - T_k}{z_j - z_k} + \sum_{k \neq j}^{L} \frac{1}{z_j - z_k} \left( \sum_{l \neq j}^{L} \frac{\mathscr{P}_{jl} - I}{z_j - z_l} - \sum_{l \neq k}^{L} \frac{\mathscr{P}_{kl} - I}{z_k - z_l} \right) \\
&\quad + \frac{1}{2} \sum_{k \neq j}^{L} \sum_{l \neq k, j}^{L} \frac{\mathscr{P}_{kl} - \mathscr{P}_{jk} - \mathscr{P}_{jl} + I}{(z_j - z_k)(z_j - z_l)} - 2 \sum_{k \neq j}^{L} \frac{\mathscr{P}_{jk} - I}{(z_j - z_k)^2} \\
&= \alpha^2 I - \sum_{k \neq j}^{L} \frac{T_j - T_k}{z_j - z_k}.
\end{aligned}
$$

$\square$

It is worth noting that similar operator identities have been noted at the level of a generalised version of (7) [28].

Proposition 1 implies that (6) is complete with respect to characterising the eigenspectra of the system.[2] The next step is to show that the variables $\{\lambda_j : j = 1, ..., L\}$ can be mapped back to Bethe roots $\{\nu_k : k = 1, ..., M\}$. This will be achieved by first solving a suitable system of linear equations.

Consider the system of $L$ linear equations for $L + 1$ parameters $q_l$:

$$
\sum_{l=0}^{L} A_{jl} q_l = 0, \qquad j = 1, ..., L, \tag{9}
$$

where

$$
A_{jl} = (\lambda_j - \alpha) z_j^l + l z_j^{l-1}.
$$

The justification for choosing this particular system will be provided at the end of the paragraph. Because the system is under-determined, there necessarily exists a non-trivial solution. Setting

$$
Q(u) = \sum_{l=0}^{L} q_l u^l
$$

allows (9) to be expressed as

$$
(\lambda_j - \alpha) Q(z_j) + Q'(z_j) = 0, \qquad j = 1, ..., L. \tag{10}
$$

In this manner a polynomial $Q(u)$ is associated to each eigenspace $W_\lambda$. If $Q(z_j) \neq 0$, then (10) can be further manipulated to acquire the form

$$
\lambda_j = \alpha - \frac{Q'(z_j)}{Q(z_j)}, \tag{11}
$$

---

[2]The claim that (6) is complete does not exclude the possibility that there are also spurious solutions of (6).

which has similar structure as (2), up to the inclusion of root multiplicities. This last observation provides the motivation to begin with the equations (9).

**Proposition 2**: Let

$$P(u) = \prod_{j=1}^{L}(u - z_j),$$

$$Q(u) = \prod_{n=1}^{N}(u - v_n)^{\mathfrak{m}_n},$$

where $\mathfrak{m}_n$ denote the root multiplicities, so the $v_n$ are distinct. If $Q(z_j) \neq 0$ for all $j = 1, ..., L$ then

$$Q''(u) - 2\alpha Q'(u) - \sum_{n=1}^{N} \frac{\mathfrak{m}_n Q(u)}{u - v_n} \frac{P'(v_n)}{P(v_n)} = 0. \tag{12}$$

*Proof.* From (6) and (11)

$$
\begin{aligned}
\lambda_j^2 &= \alpha^2 - \sum_{k \neq j}^{L} \frac{1}{z_j - z_k}\left(\lambda_j - \lambda_k\right) \\
&= \alpha^2 + \sum_{k \neq j}^{L} \frac{1}{z_j - z_k} \sum_{n=1}^{N}\left(\frac{\mathfrak{m}_n}{z_j - v_n} - \frac{\mathfrak{m}_n}{z_k - v_n}\right) \\
&= \alpha^2 + \sum_{n=1}^{N}\sum_{k=1}^{L} \frac{\mathfrak{m}_n}{(z_k - v_n)(v_n - z_j)} + \sum_{n=1}^{N} \frac{\mathfrak{m}_n}{(v_n - z_j)^2} \\
&= \alpha^2 + \sum_{n=1}^{N} \frac{\mathfrak{m}_n}{z_j - v_n} \frac{P'(v_n)}{P(v_n)} + \sum_{n=1}^{N} \frac{\mathfrak{m}_n}{(v_n - z_j)^2},
\end{aligned} \tag{13}
$$

noting that $P(v_n) \neq 0$ for all $n = 1, ..., N$ since $Q(z_j) \neq 0$ for all $j = 1, ..., L$. On the other hand using only (11)

$$
\begin{aligned}
\lambda_j^2 &= \alpha^2 - 2\alpha\frac{Q'(z_j)}{Q(z_j)} + \left(\frac{Q'(z_j)}{Q(z_j)}\right)^2 \\
&= \alpha^2 - 2\alpha\frac{Q'(z_j)}{Q(z_j)} - \left(\frac{Q'(u)}{Q(u)}\right)'\Bigg|_{u=z_j} + \frac{Q''(z_j)}{Q(z_j)} \\
&= \alpha^2 - 2\alpha\frac{Q'(z_j)}{Q(z_j)} + \sum_{n=1}^{N} \frac{\mathfrak{m}_n}{(z_j - v_n)^2} + \frac{Q''(z_j)}{Q(z_j)}.
\end{aligned} \tag{14}
$$

For equality of the expressions (13) and (14) it is required that

$$\sum_{n=1}^{N} \frac{\mathfrak{m}_n}{z_j - v_n} \frac{P'(v_n)}{P(v_n)} = \frac{Q''(z_j)}{Q(z_j)} - 2\alpha\frac{Q'(z_j)}{Q(z_j)}, \qquad j = 1, ..., L. \tag{15}$$

Set

$$R(u) = Q''(u) - 2\alpha Q'(u) - \sum_{n=1}^{N} \frac{\mathfrak{m}_n Q(u)}{u - v_n} \frac{P'(v_n)}{P(v_n)} \tag{16}$$

which is a polynomial of order less than $L$, since $Q(u)$ is at most of order $L$. It then follows from (15) that $R(u) = 0$, so (12) holds. $\qquad \square$

**Corollary 3**: The root multiplicities are $\mathfrak{m}_n = 1$ for all $n = 1, ..., N$.

Supposing $\mathfrak{m}_n > 1$, and differentiating Eq. (12) $(\mathfrak{m}_n - 2)$ times, implies that $Q^{(\mathfrak{m}_n)}(v_n) = 0$. This is a contradiction, so $\mathfrak{m}_n = 1$ for all $n = 1, ..., N$.

**Corollary 4**: The Bethe Ansatz equations (3) hold, and $N = M$.

The result follows from evaluating $R(v_n) = 0$ through (16).

## 2.1 A generalisation

If, for arbitrary $\beta \in \mathbb{R}$, the system

$$(\lambda_j - \beta)Q(z_j) + Q'(z_j) = 0, \qquad j = 1, ..., L$$

is considered instead of (10) then the above procedure can be repeated to show that now

$$Q''(u) - 2\beta Q'(u) + \left(\beta^2 - \alpha^2 - \sum_{n=1}^{N} \frac{\mathfrak{m}_n}{u - v_n} \frac{P'(v_n)}{P(v_n)}\right)Q(u) = (\beta^2 - \alpha^2)P(u)$$

as a generalisation of (12). The generalised Bethe Ansatz equations read

$$(Q''(v_n) - 2\beta Q'(v_n))P(v_n) - \mathfrak{m}_n Q'(v_n)P'(v_n) = (\beta^2 - \alpha^2)[P(v_n)]^2, \qquad n = 1, ..., N. \quad (17)$$

Richardson-Gaudin Bethe Ansatz equations of this generalised type, where the coefficient of $[P(v_n)]^2$ on the right-hand side of (17) is non-zero, have appeared through a number of approaches including the use of reflection equations methods [29, 30], non-standard pseudo-vacuua [28, 31], and the classical Yang-Baxter equation for higher spin systems [32]. The equations (17) are equivalent to those in [33] obtained from the non-standard pseudo-vacuua approach, with $N = L$ and $\mathfrak{m}_n = 1$ for all $n = 1, ..., L$.

## 3 Constructing eigenstates

Above it was shown that the Bethe Ansatz equations (3) follow, under the condition that $Q(z_j) \neq 0$ for all $j = 1, .., L$, from the operator identities (7). The next issue to address is whether the Bethe states (5) are complete. Given a solution of (3), it is possible that substitution of the roots into (5) produces a null vector. However, it is also possible that a non-null vector can be obtained by the inclusion of an appropriate normalisation factor into (5). If this is possible, then the Bethe state is said to be *regularisable*. If it is not possible, or the substitution of the roots of (3) into (5) simply does not produce an eigenvector, then the solution is said to be *spurious*. Another possibility is simply that not all eigenstates follow the operator product Ansatz adopted in (5). Taking all these factors into consideration, the remaining objective is determine under which conditions it can be proved that (5) provides a complete set of eigenstates.

Here it will prove advantageous to work with a generalised version of the operators (1). Let $\gamma \in \mathbb{C}$ and set

$$U = I - \gamma\sigma^+$$
$$\mathcal{U} = U_1 U_2, ..., U_L,$$
$$\mathcal{T}_j = \mathcal{U} T_j \mathcal{U}^{-1}$$
$$= \alpha(\sigma_j^z + 2\gamma\sigma_j^+) + \sum_{k \neq j}^{L} \frac{\mathcal{P}_{jk} - I}{z_j - z_k}.$$

The Bethe Ansatz equations (3), and the eigenvalue expressions (2), still hold for the set $\{\mathcal{T}_j : j = 1,...,L\}$, but the Bethe state expression (5) needs to be modified. Rather than simply expressing the transformed Bethe states through conjugation by $\mathcal{U}$, it is more useful to generalise the algebraic Bethe Ansatz procedure. The approach is similar to that of [34], the steps for which are outlined below.

Define

$$t_1^1(u) = \alpha I + \frac{1}{2}\sum_{j=1}^{L}\frac{1}{u-z_j}(I+\sigma_j^z),$$

$$t_2^1(u) = \sum_{j=1}^{L}\frac{1}{u-z_j}\sigma_j^+,$$

$$t_1^2(u) = 2\alpha\gamma I + \sum_{j=1}^{L}\frac{1}{u-z_j}\sigma_j^-,$$

$$t_2^2(u) = -\alpha I + \frac{1}{2}\sum_{j=1}^{L}\frac{1}{u-z_j}(I-\sigma_j^z),$$

which can be shown to satisfy the commutation relations

$$\left[t_j^i(u), t_l^k(v)\right] = \left[t_j^i(v), t_l^k(u)\right]$$

$$= \frac{\delta_j^k}{u-v}\left(t_l^i(v)-t_l^i(u)\right) + \frac{\delta_l^i}{u-v}\left(t_j^k(u)-t_j^k(v)\right), \tag{18}$$

where $\delta_j^k$ denote the Kronecker delta. It follows from the commutation relations (18) that the *transfer matrix*

$$T(u) = t_1^1(u)t_1^1(u) + t_2^1(u)t_1^2(u) + t_1^2(u)t_2^1(u) + t_2^2(u)t_2^2(u)$$

forms a commuting family:

$$[T(u), T(v)] = 0 \qquad \forall u,v \in \mathbb{C}.$$

Indeed it can be shown that

$$T(u) = \left(2\alpha^2 + \left(\sum_{j=1}^{L}\frac{1}{u-z_j}\right)^2 + \sum_{j=1}^{L}\frac{1}{(u-z_j)^2}\right)I + 2\sum_{j=1}^{L}\frac{1}{u-z_j}\mathcal{T}_j. \tag{19}$$

Using the properties

$$t_1^1(u)|0\rangle = \left(\alpha + \sum_{j=1}^{L}\frac{1}{u-z_j}\right)|0\rangle,$$

$$t_2^1(u)|0\rangle = 0,$$

$$t_2^2(u)|0\rangle = -\alpha|0\rangle,$$

and the relation

$$\left[t_2^1(u), t_1^2(u)\right] = \sum_{j=1}^{L}\frac{1}{(u-z_j)^2}\sigma_j^z,$$

it is seen that the pseudo-vacuum state is an eigenstate of the transfer matrix with eigenvalue

$$\rho(u) = 2\alpha^2 + 2\alpha \sum_{j=1}^{L} \frac{1}{u - z_j} + \left( \sum_{j=1}^{L} \frac{1}{u - z_j} \right)^2 + \sum_{j=1}^{L} \frac{1}{(u - z_j)^2}.$$

Set

$$|\Psi\rangle = \prod_{n=1}^{M} t_1^2(v_n)|0\rangle,$$

$$|\Psi_m\rangle = \prod_{n \neq m}^{M} t_1^2(v_n)|0\rangle.$$

In the algebraic Bethe Ansatz approach the following action is considered [34]

$$T(u)|\Psi\rangle = \sum_{m=1}^{M} t_1^2(v_1)...\left[ T(u), t_1^2(v_m) \right]...t_1^2(v_M)|0\rangle + \rho(u)|\Psi\rangle. \tag{20}$$

Letting $h(u) = t_1^1(u) - t_2^2(u)$ and using the key commutation relations

$$[h(u), t_1^2(v)] = \frac{2}{u - v} \left( t_1^2(u) - t_1^2(v) \right),$$

$$[T(u), t_1^2(v)] = \frac{2}{u - v} \left( t_1^2(u)h(v) - t_1^2(v)h(u) \right)$$

it is found that

$$T(u)|\Psi\rangle = \lambda(u)|\Psi\rangle + 2t_1^2(u) \sum_{m=1}^{M} \frac{\Gamma_m(v_m)}{u - v_m} |\Psi_m\rangle, \tag{21}$$

where

$$\lambda(u) = \rho(u) - 2 \sum_{m=1}^{M} \frac{\Gamma_m(u)}{u - v_m}, \tag{22}$$

$$\Gamma_m(u) = 2\alpha + \sum_{j=1}^{L} \frac{1}{u - z_j} - \sum_{n \neq m}^{M} \frac{2}{v_m - v_n}. \tag{23}$$

It needs to be emphasised that, in arriving at (22,23), it has been assumed that the parameters $\{v_m : m = 1, ..., M\}$ are pairwise distinct.

For $\gamma \neq 0$ observe also that $|\Psi\rangle$ is a non-null vector, since the projection of $|\Psi\rangle$ onto $|0\rangle$ is non-null. Moreover $|\Psi\rangle$ is an eigenvector of $T(u)$ whenever the Bethe Ansatz equations (3) hold, since $\Gamma_m(v_m) = 0$. In such a case

$$\lambda(u) = \rho(u) - 2 \sum_{j=1}^{L} \sum_{n=1}^{M} \frac{1}{(u - z_j)(z_j - v_n)}. \tag{24}$$

Using the expansion (19) it can then be checked from (24) that the eigenvalues of $\{\mathcal{T}_j : j = 1, ..., L\}$ are given by (2). Note that Eqs. (22,23) are independent of $\gamma$, and therefore also hold as $\gamma \to 0$.

**Lemma 5**: For any solution of the Bethe Ansatz equations (3), if the state $|\Phi\rangle$ is null it is nonetheless regularisable.

The result follows since

$$|\Phi\rangle = \lim_{\gamma \to 0} |\Psi\rangle$$

and because $|\Psi\rangle$ is non-null for all $\gamma \neq 0$,

$$|\overline{\Phi}\rangle = \lim_{\gamma \to 0} \frac{1}{|||\Psi\rangle||} |\Psi\rangle$$

must converge to a non-null vector. Thus $|\overline{\Phi}\rangle$ is a regularisation of $|\Phi\rangle$.

## 4 Completeness

Now attention turns to the determination of conditions under which completeness can be deduced.

**Definition 6**: The parameters $\alpha \in \mathbb{R}$ and pairwise-distinct real parameters $\{z_j : j = 1, ..., L\}$ are said to be in *general position* if

  (i) all eigenspaces $W_\lambda$ are one-dimensional;

  (ii) for all eigenspaces $W_\lambda$ the associated function $Q(u)$ has the property $Q(z_j) \neq 0$ for all $j = 1, ..., L$.

It is important to establish that the definition for *general position* above describes a generic setting. This can be seen to be so by considering the limit of large $|\alpha|$. Define parameters $n_j$, $j = 1, ..., L$ which may take values 0 or 1. Then the $2^L$ ordered $L$-tuples $(n_1, ..., n_L)$ are in one-to-one correspondence with the one-dimensional eigenspaces

$$W_\lambda = \mathrm{span}\left( \prod_{j=1}^{L} \left( \sigma_j^- \right)^{n_j} |0\rangle + \mathcal{O}(\alpha^{-1}) \right)$$

where

$$\lambda_j = (1 - 2n_j)\alpha + \mathcal{O}(1),$$

giving

$$M = \sum_{j=1}^{L} n_j.$$

In this large $|\alpha|$ limit it can be checked that for each eigenspace

$$Q(u) = \prod_{j=1}^{L} (u - z_j - (2\alpha)^{-1} + \mathcal{O}(\alpha^{-2}))^{n_j},$$

confirming that the parameters are in general position.

The main result can now be presented.

**Proposition 7**: For a system with parameters in general position, the Bethe Ansatz solution (3) provides a complete set of eigenstates through (5), up to regularisation. There are no spurious solutions of (3).

*Proof.* For any eigenvector $|\chi\rangle$ of a system with parameters in general position, there exists a solution of (3) due to Corollary 4. The roots of (3) are distinct due to Corollary 3. Consequently Eq. (21) with $\gamma = 0$ holds where $\Gamma_m(\nu_m) = 0$ for all $m = 1, ..., M$. Then either $|\Phi\rangle$ is non-null, or it is regularisable as a result of Lemma 5. In either case $|\chi\rangle$ is recovered up to normalisation, due to the eigenspaces being one-dimensional. Since this process applies to every eigenspace there exists a basis of states of the form (5) up to regularisation. There can be no spurious solutions of (3) as a consequence of Lemma 5. $\qquad\square$

It is important to confirm that there are instances where the parameters are not in general position, which has been observed in numerical solution of (3) [35–37]. Generally, roots of (3) are real-valued or arise in pairs of complex conjugates. As the system parameters are continuously varied the roots also vary, with complex-conjugate pairs transitioning to become real-valued, or vice versa. For this to happen, there must be a point at which two roots have the same value in which case the right-hand side of (3) has a singularity. The singularity is matched on the left-hand side if the coincident value of the two roots is from the set $\{z_j : j = 1, ..., L\}$, which is exactly what is observed. This behaviour is consistent with Eq. (10) , which indicates that if $Q(z_j) = 0$, then $Q'(z_j) = 0$, so the root $z_j$ is of multiplicity greater than 1. However the numerical studies also suggest that this setting is not a generic feature, and whenever it occurs the system parameters can be perturbed to resume general position.

In the case when $\alpha = 0$ the simultaneous eigenspaces are not all one-dimensional. The majority of Bethe states have the property that some Bethe roots diverge as $\alpha \to 0$, in which case they do not have the form (5) in a strict sense. It can be shown that an $su(2)$ symmetry emerges in this limit, and the Bethe states which do not have divergent Bethe roots are highest-weight states with respect to this symmetry in the limit $\alpha \to 0$. By incorporating this symmetry, methods exist to construct additional eigenstates for the system. Refer to [21, 22] for details.

## 5 Conclusion

An argument was presented to deduce that the Bethe Ansatz solution of the rational, spin-1/2 Richardson-Gaudin system is complete for generic values of the system parameters. Distinct from studies of other integrable systems, the method presented does not assume knowledge of the distribution of Bethe roots, such as in a string hypothesis, nor is it reliant on counting the number of solutions. It is instead founded on the operator identities (7). This is an approach that can be generalised. The method may be applied to a generalised version of the spin-1/2 system without $u(1)$ symmetry [30], whereby the relevant operator identities are given in [28]. Moreover, higher-spin generalisations are possible. The form of the required operator identities can be inferred from the systems of eigenvalue variable equations derived in [24, 25] for rational cases. Analyses for these generalised systems will be reported in due course.

**Funding information**    This work was supported by the Australian Research Council through Discovery Project DP150101294.

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
