# Peer review of "Completeness of the Bethe states for the rational, spin-1/2 Richardson--Gaudin system"

_SciPost Physics, doi:SciPost Phys. 3, 007 (2017)_

## Round 4 · Referee Report · Anonymous · 2017-6-14

Strengths

1.The article be well written, easy to read.
2. The method used is new and functional.
3. the model considered is simple.
4. The results are of interest to both physicists and mathematicians.

Weaknesses

1.non Weaknesses

Report

The completeness of Bethe states is a still open problem of great interest to both physicists and mathematicians. The author developed a simple method to address this problem in one of the most studied models of the research area. In my opinion, this work will be read by researchers in this area and therefore deserves to be published.

Requested changes

non Changes

---

## Round 4 · Referee Report · Anonymous · 2017-6-20

Strengths

1- Very well written and easy to follow
2- Original and simple approach to the completeness problem for a Bethe Ansatz solvable model
3- Offers great potential for generalisation to other Richardson-Gaudin systems.

Weaknesses

1 - The claim that "general position" is a generic setting appears to be the weakest part of the work, at least from a formal point of view.

They do prove it in limit of large | \alpha |, they recognise that there are instances for which it is not the case: certain specific coupling values (with matching singularities) and the generic limit \alpha=0. They appeal to numerical studies to claim that those matching singularities points are indeed non-generic. In this particular work, this appeal to numerics seems completely sufficient to validate this point, but this could become problematic for generalisations of this work, which might then require case by case numerics.

2- It focuses on a single model while possible generalisations are mentioned throughout the paper: section 2.1 mentions a generalised problem on which no further claims seem to be made.

Report

The author presents an interesting approach to the completeness problem of the Bethe ansatz for spin 1/2 Richardson-Gaudin models. By avoiding having to discuss explicitly Bethe roots configurations or explicit solution counting, the resulting proof makes it extendable to other integrable models of the Richardson-Gaudin class.

The submission deals with the lack of spurious solutions, the capacity to regularise potential "null" eignevectors, and uses the fact that the problem has a simple spectrum (one-dimensional eigenspaces) in relation with the operator identities found in proposition 1, which links together the conserved charges of this integrable model. Therefore, it present a complete proof of the completness of the BEthe Ansatz for generic parameters of the system

The results are presented very clearly and in a way which makes this work accessible to both the mathematics and physics communities.

Provided the requested minor changes are addressed, this work deserves publication in scipost.org.

Requested changes

1- The author insists on the technique not relying on string hypothesis or solution counting and claims that not much has been published in the literature concerning this problem. There is at least one reference (not mentioned by the author):

E. Mukhin, V. Tarasov and A. Varchenko, Glasgow Math. J. 51A 137 (2009) doi:10.1017/S0017089508004850 or https://arxiv.org/abs/0712.0981

which also adresses the same problem and reaches the same conclusions, it would be crucial that the author makes explicit reference to this work and adresses clearly how his work adds to and differs from theirs.

2- Following proposition 1, the author should make reference to the similar identity found in [30] (as was the case in the earlier versions of this submission (v1; v2)). Such a fact is now only acknowledged in the conclusion of this submission.

3- There is a typo in the sentence ending the proof paragraph of page 11: "The can be no spurious solutions" should read " There can be ... "

---

## Round 5 · Author Response

I thank the Editor-in-charge, and the referees, for their helpful comments on improvements.

---

## Round 5 · List of Changes

The following changes have been made in response to the Editor-in-charge:
[1.] The explanation that has been requested has been provided in the final two sentences appearing on page 3 of the revised version.
[2.] The Editor is correct, the change has been made to the equation which is now numbered (4).
[3.] Regarding equation (9) (previously (8)), explanatory text has been added before its appearance on page 5, and also preceding Proposition 2.
[4.] $t(u)$ in the equation now numbered (20) was a typographical error. It has now been corrected to read $T(u)$.
[5.] Equation numbering has been added to those unnumbered equations which were referred to in the Editorial Recommendation.
The following changes have been made in response to the Anonymous Report 169:
[1.] It was requested that the publication titled {\it On the separation of variables and completeness of the Bethe Ansatz for quantum ${\mathfrak gl}_{N}$ Gaudin model} by Mukhin, Tarasov, and Varchenko is cited and discussed. The paper by Mukhn et al. deals with a Gaudin model, as opposed to a Richardson-Gaudin model. That is, it is dealing with the $\alpha=0$ case which can be seen from the Bethe Ansatz equations which appear on page 141 of the published version, or page 5 of the arXiv version. This publication has been cited as [23] and referred in the text, within the second last paragraph preceding Section 2, on page 4.
[2.] Reference to the identity of [28] (previously [30]) has been added at the end of the proof for Proposition 1.
[3.] The typographical error referred to on page 11 has been corrected.
There were no changes requested in the Anonymous Report 165.
[1.] The explanation that has been requested has been provided in the final two sentences appearing on page 3 of the revised version.
[2.] The Editor is correct, the change has been made to the equation which is now numbered (4).
[3.] Regarding equation (9) (previously (8)), explanatory text has been added before its appearance on page 5, and also preceding Proposition 2.
[4.] $t(u)$ in the equation now numbered (20) was a typographical error. It has now been corrected to read $T(u)$.
[5.] Equation numbering has been added to those unnumbered equations which were referred to in the Editorial Recommendation.
The following changes have been made in response to the Anonymous Report 169:
[1.] It was requested that the publication titled {\it On the separation of variables and completeness of the Bethe Ansatz for quantum ${\mathfrak gl}_{N}$ Gaudin model} by Mukhin, Tarasov, and Varchenko is cited and discussed. The paper by Mukhn et al. deals with a Gaudin model, as opposed to a Richardson-Gaudin model. That is, it is dealing with the $\alpha=0$ case which can be seen from the Bethe Ansatz equations which appear on page 141 of the published version, or page 5 of the arXiv version. This publication has been cited as [23] and referred in the text, within the second last paragraph preceding Section 2, on page 4.
[2.] Reference to the identity of [28] (previously [30]) has been added at the end of the proof for Proposition 1.
[3.] The typographical error referred to on page 11 has been corrected.
There were no changes requested in the Anonymous Report 165.

---

## Editorial Decision

published